# Is There a Role for the Neutrophil-to-Lymphocyte Ratio for Rebleeding and Mortality Risk Prediction in Acute Variceal Bleeding? A Comparative 5-Year Retrospective Study

**DOI:** 10.3390/diseases13080265

**Published:** 2025-08-16

**Authors:** Sergiu Marian Cazacu, Dragos Ovidiu Alexandru, Alexandru Valentin Popescu, Petrica Popa, Ion Rogoveanu, Vlad Florin Iovanescu

**Affiliations:** 1Gastroenterology Department, University of Medicine and Pharmacy Craiova, Emergency Clinical Hospital Craiova, 200642 Craiova, Romania; sergiu.cazacu@umfcv.ro (S.M.C.); ion.rogoveanu@umfcv.ro (I.R.); vlad.iovanescu@umfcv.ro (V.F.I.); 2Biostatistics Department, University of Medicine and Pharmacy Craiova, Emergency Clinical Hospital Craiova, 200642 Craiova, Romania; dragos.alexandru@umfcv.ro; 3Gastroenterology Department, Emergency Clinical Hospital Craiova, 200642 Craiova, Romania; popescualexandruvalentin97@yahoo.com

**Keywords:** acute variceal bleeding, upper gastrointestinal bleeding, prognostic scores, Child-Turcotte-Pugh score, MELD, neutrophil-to-lymphocyte ratio

## Abstract

(1) Background: Acute variceal bleeding (AVB) represents an important cause of upper gastrointestinal bleeding (UGIB). Several prognostic scores may be useful for assessing mortality and rebleeding risk, with the Glasgow-Blatchford score (GBS) and Rockall score being the most commonly used for non-variceal bleeding. Scores assessing liver failure (MELD and Child) do not reflect bleeding severity. The neutrophil-to-lymphocyte ratio (NLR) increases in UGIB and can predict survival and rebleeding. (2) Methods: We analyzed the predictive role of NLR, GBS, Rockall, AIMS65, Child, and MELD for mortality (48 h, 5-day, in-hospital, and 6-week) and rebleeding in AVB patients admitted to our hospital from 2017 to 2021. ROC analysis was performed, and a multivariate analysis with logistic regression was used to construct a simplified model. (3) Results: A total of 415 patients were admitted. NLR exhibited fair accuracy for 48-h mortality (AUC 0.718, 95% CI 0.597–0.839, *p* < 0.0001), with limited predictive value for medium-term mortality. The NLR accuracy was better than that of the GBS and Rockall score, similar to that of the AIMS65 and Child scores, but inferior to that of MELD. The value for all scores in predicting rebleeding was poor, with the highest AUC for the NLR. (4) Conclusions: The NLR exhibited reasonable accuracy in predicting short-term mortality in AVB. Our model (including NLR, age, creatinine, bilirubin, albumin, INR, platelet count, HCC, and etiology) demonstrated 80.72% accuracy in predicting 6-week mortality.

## 1. Introduction

Acute variceal bleeding (AVB) represents a major complication of portal hypertension and a leading cause of upper gastrointestinal bleeding (UGIB). Mortality is estimated at approximately 15–20% [1,2] and is often correlated with the severity of the disease, most frequently assessed based on Child–Pugh and MELD scores [3]. Treatment of AVB includes band variceal ligation for esophageal variceal and type I gastroesophageal varices (GOV) bleeding and obliteration with cyanoacrylate for type II GOV and isolated gastric varices, together with vasoactive treatment (preferably initiated before endoscopy) and prophylactic antibiotherapy [4,5,6]. A Sengstaken–Blakemore tube can be of temporary utility in cases of endoscopic failure. Salvage TIPS (Transjugular Intrahepatic Porto-systemic Shunt) or stent placement may be used in refractory AVB, and current guidelines recommend early TIPS in patients with Child scores B or C and active variceal bleeding at the time of endoscopy [3]. Despite these recommendations, access to TIPS varies across different countries.

Several prognostic scores have been proposed for predicting mortality, rebleeding, the need for intervention, and intensive care unit admissions [7]. The Glasgow–Blatchford score (GBS) and Rockall scores (clinical and full) are the most widely used scores for non-variceal bleeding; a more effective score for cirrhotic patients with bleeding is the AIMS65, proposed by Saltzmann in 2011, because it includes items such as albumin level, INR, and altered consciousness [8]. However, AVB prognosis is more often related to the severity of liver failure (reflected by the values of Child–Pugh–Turcotte or MELD score); the authors of the majority of studies have concluded that the accuracy of classical scores in variceal bleeding is inferior compared to that in non-variceal bleeding. A meta-analysis including 28 studies found that the Child–Turcotte–Pugh (CTP) score provides greater accuracy than classical prognostic scores (AIM65, GBS, and Rockall score) [9], with the MELD score noted as superior to the CTP score in some studies [10,11,12,13]. Most classical scores (excluding AIM65) do not include parameters associated with liver failure, which may alter the accuracy of AVB assessment. Some variants of the CTP and MELD scores (such as creatinine-CTP, MELD-3, MELD-Na, and UKELD) can be used for risk assessment [14]. A general prognostic score for assessing patients with UGIB is represented by the Charlson comorbidity index (CCI), which includes the presence of associated comorbidities and advanced age [15]. A major limitation of prognostic scores in AVB is the fact that classical prognostic scores reflect only bleeding severity (excluding albumin level in AIMS65), whereas liver failure scores (CTP and MELD) are not correlated with bleeding severity. Several studies assessing the accuracy of prognostic scores are summarized in Appendix A.

The neutrophil-to-lymphocyte ratio (NLR) reflects an altered ratio between neutrophils (which stimulate inflammation via several cytokines) and lymphocytes (which correlate with immunoregulatory mechanisms) [16,17,18]. The authors of several published studies and meta-analyses have evaluated the importance of NLR in the outcome prediction of patients with acute pancreatitis, severe burns, ischemic and hemorrhagic stroke, myocardial infarction, venous thromboembolism, sepsis, COVID-19 pneumonia, and cancer [16,19,20,21,22,23,24,25,26,27,28,29,30,31], in addition to the risk of metabolic syndrome and MAFLD [32,33], cardiac surgery complications [34], encephalopathy [35], TIPS and virus C-cirrhosis complications [36,37]. Most studies regarding NLR and bleeding focus on intracerebral hemorrhage; in the acute phase, an increase in total leucocyte, monocyte, and granulocyte counts is recorded, combined with a reduction in lymphocyte count, and negative regulatory T cell levels decrease in the early phase of intracerebral bleeding, suggesting impaired control of inflammation by the immune system. High NLR levels are associated with gastrointestinal bleeding risk in patients with basal ganglia hemorrhage [22]. In patients with an unfavorable prognosis, CD3+ and CD3+CD4+ T lymphocyte counts are lower at an early stage following intracerebral hemorrhage [23]. Gastrointestinal bleeding can cause an inflammatory response, and peripheral blood inflammation may be associated with disease outcome [18]. Persistent lymphopenia appears more frequently in acute-onset diseases and may be induced by progressive inflammation; in comparison, increased neutrophil counts may be noted in acute bleeding and SIRS [16,17]. NLR levels can be increased through the use of corticosteroids (which increase neutrophil levels and may decrease lymphocyte counts); in comparison, dehydration can decrease both neutrophil function and lymphocyte counts. Thus far, only a few research groups have assessed the predictive role of NLR in mortality and rebleeding in acute gastrointestinal bleeding [17,18,24,38,39], with higher NLRs associated with mortality risk [17,18] and rebleeding risk [24].

The objective of our study was to evaluate the prognostic role of NLR as compared to classical scores (GBS and modified GBS, CRS, FRS, AIMS65, Charlson comorbidity index, or CI), liver failure scores (CTP, CTP-creatinine, MELD, MELD-3, and UKELD), and several new scores (ALBi and PALBi) for assessing mortality and early rebleeding risk.

## 2. Materials and Methods

We performed a retrospective study including all patients with acute variceal bleeding and cirrhosis admitted to the Emergency Clinical Hospital Craiova between 1 January 2017, and 31 December 2021. We collected data from the hospital’s Hippocrate computerized system, and the analysis of medical records was subsequently performed to exclude patients with non-variceal or lower gastrointestinal bleeding. We searched all patients aged over 16 years admitted for hematemesis, melena, gastrointestinal bleeding, or esophageal varices with bleeding (ICD-10 codes K92.0, K92.1, K92.2, and I85.0) with concomitant liver cirrhosis or chronic liver failure (ICD-10 codes K70, K72, K74). The results of laboratory analyses completed upon admission were collected. Inclusion criteria comprised acute variceal bleeding confirmed by endoscopy in patients with cirrhosis, confirmed by typical clinical and laboratory data and ultrasound results. Patients aged younger than 16 years, with non-cirrhotic variceal bleeding, with BCLC stage D hepatocellular carcinoma or stage IV carcinomas in other locations (due to high non-bleeding related mortality), and those with missing data were excluded. All patients signed an informed consent form for the use of their personal data. The protocol was approved by the Local Ethics Committee (approval number 11977; 24 March 2020).

The standard therapy for AVB included Terlipressin treatment for 3–5 days, saline and glucose solutions, 80 mg/day proton pump inhibitors, prophylactic antibiotherapy, and correction of clotting disorders in selected cases; cases exhibiting hemodynamic instability were managed through the use of a Sengstaken-Blakemore tube before endoscopy. Blood transfusions were recommended with a target hemoglobin value of 8 g/dl. Endoscopy was performed during the first 24 h of admission; variceal band ligation was used for esophageal varices or GOV1 (gastroesophageal varices). Cases with immediate failure were treated with Sengstaken-Blakemore tube placement, with endoscopy repeated during the first 24 h. For large varices or those with red signs, no active bleeding, and no white nipple, variceal band ligation was also performed if no other lesions were found upon endoscopy. Following endoscopy, all patients were closely monitored for continued or early recurrent bleeding (new onset of hematemesis, hematochezia, or melena accompanied by hemodynamic instability or hemoglobin decline of more than 3 g/dl) [5,6]; a new endoscopic hemostatic procedure is indicated in cases of recurrent bleeding. In cases with persistent bleeding, the Sengstaken-Blakemore tube was used.

The main outcomes in patients with acute variceal bleeding were early mortality, 6-week mortality, and early rebleeding. Because some patients with variceal bleeding can develop extensive bleeding accompanied by early death, 48 h, 5-day, and in-hospital mortality were assessed in our study, combined with 6-week mortality and early rebleeding; classical prognostic scores (CRS, FRS, GBS, modified GBS, and AIMS65), CTP and MELD scores, new scores (ALBi and PALBi) and the NLR value were analyzed to assess their predictive value (Appendix A) [10,13,40,41]. For MELD, we used the formula MELD = 9.57 * ln (Serum Cr) + 3.78 * ln (Serum Bilirubin) + 11.20 * ln (INR) + 6.43, rounded to the nearest integer [10,13,14,40,41].

Statistical analyses were performed using Microsoft Excel (Microsoft Corp., Redmond, WA, USA), with XLSTAT 2016 add-on for MS Excel (Addinsoft SARL, Paris, France) and IBM SPSS Statistics 20.0 (IBM Corporation, Armonk, NY, USA). Descriptive data were generated for the patients’ characteristics, including percentages for categorical variables and means with standard deviations for continuous variables. For continuous variables, the Mann–Whitney test was used to assess the differences between groups, while differences regarding proportions between groups were evaluated using the Chi-square test. We performed univariate and multivariate logistic analyses for factors associated with mortality, including, in the multivariate model, factors identified through univariate logistic regression analysis with *p* < 0.2.

## 3. Results

A total of 415 patients with AVB were included in the analysis; the median age was 58.2 ± 10.9 years, and nearly 2/3 were male. Esophageal varices caused 92.1% of AVB cases; thirty-two cases of bleeding gastric varices and one case of bleeding jejunal varices were noted. Most patients were classed as Child B (54.7%) and C (26.1%). Alcoholic etiology was the most frequent cause (77.8%). The most frequent complication was hepatic encephalopathy (20.7%); hepatocellular carcinoma and portal vein thrombosis were noted in 5.1% and 3.9% of patients, respectively. In-hospital mortality for all patients was noted as 15.7%, and 6-week mortality was noted as 23.1%; for Child C class patients, in-hospital and 6-week mortality were noted as 27.2% and 41.7%. The early rebleeding rate in our cohort was 8.1% (Table 1).

The accuracy of prognostic scores was evaluated by constructing the area under the curve (AUC) and 95% confidence interval (95% CI) for mortality prediction (Table 2 and Table 3, Figure 1A–H).

For 48 h mortality prediction, the most accurate scores were MELD-3 (AUC 0.808, 95% CI 0.694–0.922), MELD (AUC 0.807, 95% CI 0.697–0.916), MELD-Na (AUC 0.793, 0.676–0.910), and ALBi (AUC 0.792, 95% CI 0.680–0.903), followed by NLR (AUC 0.718, 95% CI 0.597–0.839), CTP-creatinine (AUC 0.693, 95% CI 0.558–0.828), PALBi (AUC 0.692, 95% CI 0.569–0.815), CTP (AUC 0.689, 95% CI 0.554–0.824), AIMS65 (AUC 0.673, 95% CI 0.546–0.800), and UKELD (AUC 0.671, 95% CI 0.541–0.801); all other classical prognostic scores, excluding AIMS65, exhibited low accuracy (AUC below 0.61 for GBS, mGBS, CRS, and FRS). For 5-day mortality, a similar classification of predictive value but with lower accuracy was noted; the most accurate scores were MELD (AUC 0.769, 95% CI 0.680–0.859), MELD-3 (AUC 0.762, 95% CI 0.667–0.857), ALBi (AUC 0.752, 95% CI 0.661–0.843), and MELD-Na (AUC 0.749, 95% CI 0.653–0.845), followed by CTP-creatinine (AUC 0.701, 95% CI 0.600–0.803), CTP (AUC 0.701, 95% CI 0.59–0.799), NLR (AUC 0.672, 95% CI 0.574–0.770), UKELD (AUC 0.671, 95% CI 0.569–0.773), AIMS65 (AUC 0.654, 95% CI 0.555–0.752), and PALBi (AUC 0.650, 95% CI 0.553–0.747). The same trend was noted for in-hospital and 6-week mortality; for in-hospital mortality, the most accurate scores were ALBi, MELD-Na, MELD-3, MELD, AIMS65, CTP-creatinine, CTP, NLR, and PALBi; in comparison, for 6-week mortality, the most accurate scores were ALBi (AUC 0.708, 95% CI 0.646–0.769) and MELD-Na (AUC 0.700, 95% CI 0.637–0.764), followed by MELD-3 (AUC 0.692, 95% CI 0.628–0.757), CTP-creatinine (AUC 0.682, 95% CI 0.617–0.747), UKELD (AUC 0.672, 95% CI 0.607–0.737), MELD (AUC 0.666, 95% CI 0.602–0.730), CTP (AUC 0.652, 95% CI 0.585–0.718), AIMS65 (AUC 0.643, 95% CI 0.578–0.708), and NLR (AUC 0.619, 95% CI 0.554–0.684); classical scores exhibited a lower accuracy (AUC below 0.61), although statistical significance was attained for the modified GBS, GBS, and FRS.

For NLR, the cutoff values, sensitivity, specificity, and accuracy were 5.875, 82.6%, 62.7%, and 63.8% for 48 h mortality, 4.574, 77%, 49.6%, and 53.6% for in-hospital mortality, and 4, 80.6%, 45.3%, and 54.1% for 6-week mortality. All scores demonstrated poor predictive value for early rebleeding, with none having an AUC above 0.6 (Table 4, Figure 2A,B).

We performed univariate and multivariate analyses of clinical and laboratory parameters associated with 48 h and 6-week mortality (Table 5 and Table 6), followed by logistic regression to obtain the most accurate model.

Next, we constructed new models for predicting 48-h and 6-week mortality based on the variables with *p* < 0.2 in univariate logistic regression models, including the NLR and categorical and numerical parameters, but excluding scores such as MELD, CTP, ALBi, or PALBI. A final model for mortality prediction was subsequently obtained for 6-week mortality, with sufficient accuracy and specificity and lower sensitivity, but fewer factors, and fairly good correctness; for 48-h mortality, the sensitivity was good; however, the specificity was poor (Table 7, Table 8, Table 9 and Table 10).

## 4. Discussion

NLR is associated with the severity of inflammation and may identify cirrhotic patients at increased risk of mortality and readmissions for hepatic encephalopathy [35]. It has also been shown to have a positive correlation with CTP score and be of independent predictive value for survival [35,37]. Because NLR is correlated with the level of inflammation, it may increase in parallel with the severity of encephalopathy symptoms by increasing the neurotoxicity induced by ammonia through the blood–brain barrier [35]. Both higher neutrophil count and lower lymphocyte count were noted in decompensated cirrhotic patients compared to those with compensated disease [35]. The NLR can increase in patients with gastrointestinal bleeding because of associated inflammation and high neutrophil counts [17,18]. NLR values may be influenced by age, the etiology of cirrhosis, and the presence of diabetes [37], thus complicating the search for an ideal cutoff [37]. In our study, the NLR demonstrated fairly good prognostic value for 48 h mortality; however, the simplified model primarily predicted 48 h survival. The NLR predictive value decreased from short-term to medium-term survival (AUC 0.718, 95% CI 0.597–0.839 for 48 h mortality, 0.672, 95% CI 0.574–0.770 for 5-day mortality, 0.631, 95% CI 0.551–0.711 for in-hospital mortality, and 0.619, 95% CI 0.554–0.684 for 6-week mortality). The accuracy of NLR was superior to that of classical prognostic factors (GBS, mGBS, CRS, and FRS) and similar to AIMS65, with the AUC for NLR slightly superior to AIMS65 for both 48 h and 5-day mortality but marginally inferior for in-hospital and 6-week mortality. In a study on patients with both non-variceal and AVB, the AUC for in-hospital mortality was 0.640 for NLR, 0.662 for GBS, 0.747 for FRS, 0.687 for NLR-GBS, and 0.763 for NLR-FRS [18].

In our study, the accuracy of classical prognostic scores (clinical and full Rockall scores, GBS, and modified GBS) was poor for all analyzed outcomes (in-hospital and 6-week mortality and early rebleeding), with no significant differences between scores; in all of these cases, the AUC was below 0.6. The accuracy of classical prognostic scores in predicting mortality and other clinical outcomes in AVB is considered inferior compared to non-variceal UGIB [10,14,40,42,43] because the mortality rate in AVB is more closely related to the severity of liver failure than the severity of bleeding [44]. In several studies comparing classical scores and those evaluating liver failure, the AUC for the Rockall score ranged from 0.533 and 0.834, with values below 0.7 in [40,42,43,45,46,47], values between 0.7 and 0.8 in [10,44,45,48,49], and values above 0.8 in [50,51]. In the literature, AUC values for the Glasgow–Blatchford score between 0.56 and 0.781 have been recorded. Values below 0.7 have been noted in some studies [40,42,43,44,45,47]; in comparison, the authors of other studies have reported AUC values between 0.7 and 0.8 [46,48,50,52,53], and no study authors have reported an AUC above 0.8. For the AIMS65, the prognostic value in our study was slightly better than GBS, mGBS, CRS, and FRS, with AUC values of 0.673, 0.672, and 0.643 for 48 h, in-hospital, and 6-week mortality. In the literature, values between 0.525 and 0.97 have been reported in several studies, with values below 0.7 in [14,40,42,43,45], values between 0.7 and 0.8 in [47,50], and values above 0.8 in [10,13,46,48,49,53,54,55]. In most studies, AIMS65 (which incorporates albumin level and an indicator of liver failure) appears to be superior to the Rockall score or GBS [10,43,45,46,47,48,49,53]; few studies have demonstrated the superiority of both RS and GBS, however [42,50].

In our study, MELD, MELD-3, and MELD-Na demonstrated superior accuracy compared with CTP and CTP-creatinine scores in predicting short- and medium-term mortality. Both MELD and CTP scores were superior to classical prognostic scores and AIMS65. In most studies, CTP and MELD appear to be superior to classical scores in assessing mortality risk [10,14,40,45,52,56]; similar levels of accuracy have been reported in other studies [50]. In comparison, in two other studies, AIMS65 was superior to CTP and MELD in predicting both in-hospital and 6-week mortality [46,54] but demonstrated no differences in terms of performance compared to MELD-Na [46]. In the literature, the values of AUC for CTP score have been estimated to be between 0.668 and 0.9, with values below 0.7 reported in one study [57], values between 0.7 and 0.8 reported in some studies [9,12,45,46,49,58,59,60], and values above 0.8 reported in others [14,40,43,48,50,52,54,56,61,62,63,64,65,66]. For MELD score and variants, the AUC was estimated between 0.688 and 0.88, with values below 0.7 in [14,43,57], between 0.7 and 0.8 in [9,40,45,46,58,59,62,63,64,65,66,67,68], or above 0.8 in [10,11,12,13,48,52,54,55,56,60,61]. Direct comparison between CTP and MELD has shown conflicting results; in some studies, CTP was superior [14,40,43,51,65] or exhibited similar results [9,45,46,48,49,50,52,54,56,57,58,59,61,62,63,64,65,69]; in comparison, in other studies, CTP was proven inferior to MELD [10,11,12,13]. In a systematic review and meta-analysis, the AUC for predicting in-hospital mortality was 0.824 for CTP, 0.9793 for AIMS65, 0.788 for MELD, 0.75 for the full Rockall score, and 0.683 for GBS, with CTP having the highest sensitivity and AIMS having the highest specificity; for follow-up mortality, MELD appears to be superior to CTP, with an AUC of 0.798 for MELD, 0.77 for AIMS65, and 0.746 for CTP; in comparison, the values for full Rockall score and GBS were below 0.7 (because they did not include an assessment of liver failure) [1]. Some new proposed scores, such as ALBi (albumin–bilirubin), PALBi (platelet–albumin–bilirubin), or CAGIB (diabetes, hepatocellular carcinoma, albumin, bilirubin, and creatinine), have also been proposed in some studies [13,57,62,63,64,65,66,67,68,69,70]; in our study, ALBi, but not PALBi, demonstrated good prognostic value, close to the MELD variants.

The value of all scores in predicting early rebleeding was poor in our study, with the highest value for NLR. In one published study, NLR and PLR increased in AVB patients who suffered rebleeding; the AUC for rebleeding was 0.7037 for NLR and 0.7468 for PLR [39]. Although we did not find good predictive value for NLR in early rebleeding, the fact that NLR had the highest score for the prediction of early rebleeding seems encouraging and warrants further analysis in future studies.

In our study, the accuracy of all scores decreased for medium-term prognosis, with high AUC values of 0.700 and 0.708 for MELD-Na and ALBi only. Classical scores (Rockall and GB scores) do not include an assessment of the degree of liver failure; in comparison, CTP and MELD scores are focused only on the severity of liver failure and not the severity of bleeding. Our simplified model (NLR, age, creatinine, bilirubin, albumin, INR, platelet count, HCC presence, and etiology) can be of utility in predicting 6-week mortality, with excellent survival prediction accuracy (94.14%) but only weak mortality prediction accuracy (41.41%). In-hospital mortality prediction accuracy was poor, however, because only the survival prediction was accurate (99.46%), whereas the mortality prediction was highly inaccurate (9.09%). In the literature, some nomograms for MELD or CTP have been proposed to improve the accuracy of prognostic scores [71,72,73,74,75,76,77,78,79,80]; however, challenges may be encountered in clinical applications because of complicated formulas.

Several limitations of the current study should be noted. The small number of patients, combined with the monocentric and retrospective nature of the study, significantly limits the generalizability of our findings; larger or multicenter studies will be necessary for external validation and improved data accuracy. All scores had mild specificity, while sensitivity was high in most scores. No score was found to be highly accurate for the analyzed outcomes (excluding MELD in 48 h mortality analysis); the prediction of early rebleeding was particularly inaccurate. Early or salvage TIPS is not available at our center, which may have impacted both mortality and rebleeding. Subgroup analysis related to esophageal or gastric variceal bleeding was not possible because of the small number of gastric varices cases. Future studies analyzing the role of NLR and other simple hematological parameters for the prognosis of variceal and non-variceal bleeding, possibly in combination with other factors, may help to accurately predict both mortality and rebleeding and also more effectively manage patients with severe bleeding and hemodynamic instability.

## 5. Conclusions

In this study, the neutrophil-to-lymphocyte ratio played a fairly good prognostic role in predicting short-term mortality in patients with acute variceal bleeding, with decreasing accuracy for medium-term prognosis; its accuracy was similar to that of AIMS65 (for short- and medium-term mortality) and the CTP score (short- and medium-term mortality) but was inferior to that of MELD variants. Classical prognostic scores (Glasgow–Blatchford and Rockall) were of lower prognostic value than the NLR, AIMS65, and MELD variants. MELD scores are the most accurate for mortality prediction. No score showed particular effectiveness in predicting early rebleeding, although the NLR was noted as having the highest AUC (0.600).

## Figures and Tables

**Figure 1 diseases-13-00265-f001:**
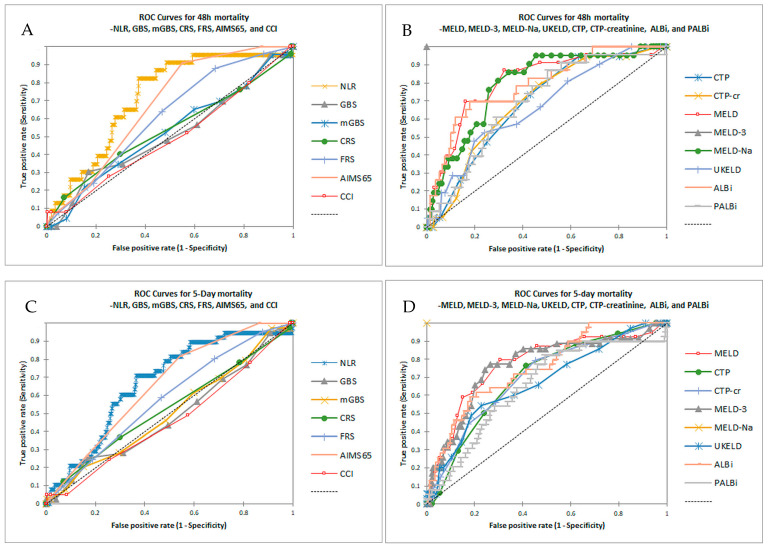
(**A**,**B**). ROC for 48 h mortality for NLR compared to GBS, mGBS, CRS, FRS, AIMS65, and CCI (**A**) and CTP, CTP-creatinine, MELD, MELD-3, MELD-Na, UKELD, ALBi, and PALBi (**B**). (**C**,**D**). ROC for 5-day mortality for NLR compared to GBS, mGBS, CRS, FRS, AIMS65, and CCI (**C**) and CTP, CTP-creatinine, MELD, MELD-3, MELD-Na, UKELD, ALBi, and PALBi (**D**). (**E**,**F**). ROC for in-hospital mortality for NLR compared to GBS, mGBS, CRS, FRS, AIMS65, and CCI (**E**) and CTP, CTP-creatinine, MELD, MELD-3, MELD-Na, UKELD, ALBi, and PALBi (**F**). (**G**,**H**). ROC for 6-week mortality for NLR compared to GBS, mGBS, CRS, FRS, AIMS65, and CCI (**G**) and CTP, CTP-creatinine, MELD, MELD-3, MELD-Na, UKELD, ALBi, and PALBi (**H**).

**Figure 2 diseases-13-00265-f002:**
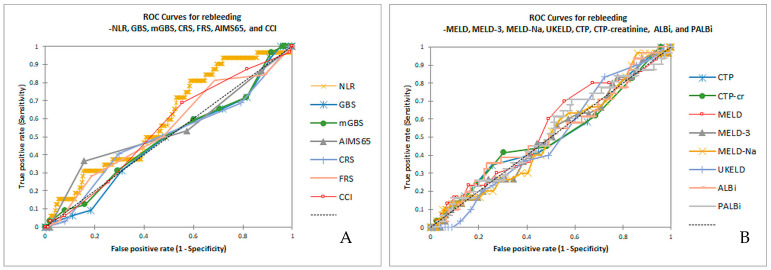
(**A**,**B**). ROC for early rebleeding prediction for NLR compared to GBS, mGBS, CRS, FRS, AIMS65, and CCI (**A**) and CTP, CTP-creatinine, MELD, MELD-3, MELD-Na, UKELD, ALBi, and PALBi (**B**).

**Table 1 diseases-13-00265-t001:** Characteristics of patients with acute variceal bleeding (N = 415).

Age (years); mean ± standard deviation (range)	58.2 ± 10.9 (28–85)
Age (n, %) < 60 60–79 >80	204 (49.2) 204 (49.2) 7 (1.7)
Gender Males/Females (% Males)	273/142 (65.8%)
Etiology (%)	
Esophageal varices	382 (92.1)
Gastric varices	32 (7.7)
Jejunal varices	1 (0.2)
Blood transfusion pts/%/mean	235/56.6/3.1 ± 1.9
Laboratory analyses median (IQR)	
Hemoglobin	8.13 (6.625–10)
Neutrophil count	6200 (3850.5–9660.5)
Lymphocyte count	1300 (800–1900)
Platelet count	94,730 (67,140–134,450)
Urea	54 (36–77)
Creatinine	0.72 (0.63–0.885)
INR	1.55 (1.34–1.88)
Albumin	2.7 (2.3–3.1)
Total bilirubin	2.11 (1.215–3.445)
ALT	35 (25–62.5)
Na	136 (132–139)
Systolic blood pressure median (IQR)	120 (100–130.5)
Child class %	
A	3.9
B	51.7
C	44.4
Etiology %	
Alcohol	323 (77.8)
Hepatitis B ± D	26 (6.2)
Hepatitis C	64 (15.4)
Biliary	2 (0.5)
Complications, comorbidities No. (%)	
Encephalopathy	86 (20.7)
Hepatocellular carcinoma	21 (5.1)
Portal vein thrombosis	16 (3.9)
Pulmonary	2 (0.5)
Cardiovascular	10 (2.4)
Diabetes	71 (17.1)
Kidney	9 (1.9)
Other neoplasia	8 (1.9)
Early rebleeding rate (%)	32 (8.1)
Mortality (%)	
48 h	25 (6)
5-day	41 (9.9)
In-hospital	65 (15.7)
6-week	96 (23.1)
Child A/B/C % (in-hospital)	0/9/22.2
Child A/B/C % (6-week)	0/19.1/33.9

**Table 2 diseases-13-00265-t002:** Main prognostic scores for 48 h and 5-day mortality.

	48 h Mortality						5-Day Mortality					
	AUC	95% CI	*p*-Value	Sn	Sp	Cutoff	AUC	95% CI	*p*-Value	Sn	Sp	Cutoff
NLR	0.718	0.597–0.839	*<0.0001*	82.6	62.7	5.875	0.672	0.574–0.770	*0.001*	71.1	63.4	5.875
GBS	0.511	0.388–0.633	0.865	47.8	51	10	0.488	0.393–0.583	0.574	43.6	50.5	10
mGBS	0.523	0.400–0.646	0.713	52.2	51.8	7	0.504	0.408–0.599	0.939	61.5	40.1	6
CRS	0.537	0.418–0.656	0.539	40	70	4	0.529	0.434–0.623	0.552	36.6	70.1	4
FRS	0.609	0.488–0.729	0.077	64	53.2	6	0.580	0.484–0.676	0.101	80.5	31.6	5
AIMS65	0.673	0.546–0.800	*0.008*	90.9	45	1	0.654	0.555–0.752	*0.002*	81.6	45.6	1
CCI	0.492	0.375–0.608	0.888	52	43.1	4	0.461	0.370–0.552	0.404	48.8	42.5	4
CTP	0.689	0.554–0.824	*0.006*	73.7	56.9	8	0.697	0.595–0.799	*<0.0001*	76.5	58.5	9
CTP-creatinine	0.693	0.558–0.828	*0.005*	78.9	53.4	9	0.701	0.600–0.803	*<0.0001*	79.4	54.8	9
MELD	0.807	0.697–0.916	*<0.0001*	87	67.7	10	0.769	0.680–0.859	*<0.0001*	79.,5	69.3	10
MELD-3	0.808	0.694–0.922	*<0.0001*	85.7	72.1	27	0.762	0.667–0.857	*<0.0001*	77.1	73.6	27
MELD-Na	0.793	0.676–0.910	*<0.0001*	85.7	65.8	28	0.749	0.653–0.845	*<0.0001*	77.1	71.9	29
UKELD	0.671	0.541–0.801	*0.010*	52.4	76	54	0.671	0.569–0.773	*0.001*	54.3	77.3	54
ALBi	0.792	0.680–0.903	*<0.0001*	69.6	82.2	−0.829	0.752	0.661–0.843	*<0.0001*	59	82	−0.843
PALBi	0.692	0.569–0.815	*0.002*	87	49.6	−2.678	0.650	0.553–0.747	*0.003*	82.1	50.7	−2.678

NLR = neutrophil-to-lymphocyte ratio; GBS = Glasgow-Blatchford score; mGBS = modified Glasgow-Blatchford score; CRS = clinical Rockall score; FRS = full Rockall score; CCI = Charlson comorbidity index; CTP = Child-Turcotte-Pugh score; CTP-creatinine = Child-Turcotte-Pugh-creatinine score; ALBi = albumin-bilirubin score; PALBi = platelet-albumin-bilirubin score; Sn = sensitivity; Sp = specificity. Statistically significant *p*-values are marked with italicized fonts.

**Table 3 diseases-13-00265-t003:** Main prognostic scores for in-hospital and 6-week mortality.

	In-Hospital Mortality						6-Week Mortality					
	AUC	95% CI	*p*-Value	Sn	Sp	Cutoff	AUC	95% CI	*p*-Value	Sn	Sp	Cutoff
NLR	0.631	0.551–0.711	*0.001*	77	49.6	4.574	0.619	0.554–0.684	*<0.0001*	80.6	45.3	4
GBS	0.562	0.482–0.641	0.129	30.6	84.4	12	0.589	0.524–0.654	*0.007*	60.6	55.1	10
mGBS	0.576	0.496–0.655	0.063	27.4	86.2	9	0.603	0.539–0.668	*0.002*	60.6	55.7	7
CRS	0.562	0.483–0.640	0.122	40.6	71.2	4	0.526	0.462–0.590	0.423	34.9	70.9	4
FRS	0.592	0.512–0.671	*0.023*	60.3	54.4	6	0.570	0.506–0.635	*0.033*	58.1	55.7	6
AIMS65	0.672	0.592–0.751	*<0.0001*	80	47.1	1	0.643	0.578–0.708	*<0.0001*	73	48.5	1
CCI	0.545	0.467–0.623	0.260	62.5	44.4	4	0.582	0.518–0.647	*0.012*	61.5	46.3	4
CTP	0.661	0.579–0.743	*<0.0001*	67.9	59.4	9	0.652	0.585–0.718	*<0.0001*	43.8	79.9	10
CTP-creatinine	0.668	0.586–0.751	*<0.0001*	69.6	55.5	9	0.682	0.617–0.747	*<0.0001*	41.7	87.6	11
MELD	0.701	0.623–0.778	*<0.0001*	67.2	70.3	10	0.666	0.602–0.730	*<0.0001*	47.5	85	12
MELD-3	0.706	0.627–0.786	*<0.0001*	61.4	76.7	28	0.692	0.628–0.757	*<0.0001*	49.5	84.1	30
MELD-Na	0.707	0.628–0.787	*<0.0001*	64.9	73	29	0.700	0.637–0.764	*<0.0001*	57.7	75.9	29
UKELD	0.665	0.583–0.747	*<0.0001*	49.1	83.6	55	0.672	0.607–0.737	*<0.0001*	41.2	85.5	55
ALBi	0.733	0.658–0.808	*<0.0001*	56.5	83.4	−0.850	0.708	0.646–0.769	*<0.0001*	49	83.8	−0.871
PALBi	0.623	0.544–0.703	*0.002*	69.4	54.7	−2.515	0.570	0.505–0.635	*0.036*	67.3	47.6	−2.781

NLR = neutrophil-to-lymphocyte ratio; GBS = Glasgow-Blatchford score; mGBS = modified Glasgow-Blatchford score; CRS = clinical Rockall score; FRS = full Rockall score; CCI = Charlson comorbidity index; CTP = Child-Turcotte-Pugh score; CTP-creatinine = Child-Turcotte-Pugh-creatinine score; ALBi = albumin-bilirubin score; PALBi = platelet-albumin-bilirubin score; Sn = sensitivity; Sp = specificity. Statistically significant *p*-values are marked with italicized fonts.

**Table 4 diseases-13-00265-t004:** Main prognostic scores for early rebleeding with sensitivity, specificity, and cutoff.

	AUC	95% CI	p-Value	Sn	Sp	Cutoff
NLR	0.600	0.492–0.707	0.069	81.3	41.3	4.050
GBS	0.475	0.372–0.578	0.635	50	51.2	10
mGBS	0.493	0.389–0.596	0.890	50	51.7	7
CRS	0.497	0.393–0.601	0.952	40.6	70.2	4
FRS	0.543	0.437–0.649	0.426	50	52.4	6
AIMS65	0.541	0.431–0.650	0.468	53.3	42.7	1
CCI	0.544	0.438–0.651	0.415	68.8	44.4	4
CTP	0.505	0.395–0.615	0.929	34.5	74.5	10
CTP-creatinine	0.505	0.396–0.615	0.925	41.4	69.7	10
MELD	0.541	0.432–0.651	0.461	70	44.2	7
MELD-3	0.502	0.394–0.610	0.969	50	49.3	21
MELD-Na	0.500	0.393–0.608	0.997	53.3	49.6	23
UKELD	0.494	0.386–0.601	0.910	60	41.1	50
ALBi	0.519	0.412–0.626	0.731	35.5	76.7	−0.870
PALBi	0.517	0.411–0.624	0.749	71	41.5	−2.860

NLR = neutrophil-to-lymphocyte ratio; GBS = Glasgow–Blatchford score; mGBS = modified Glasgow-Blatchford score; CRS = clinical Rockall score; FRS = full Rockall score; CCI = Charlson comorbidity index; CTP = Child–Turcotte–Pugh score; ALBi = albumin–Bilirubin score; PALBi = platelet–albumin–Bilirubin score; Sn = sensitivity, Sp = specificity.

**Table 5 diseases-13-00265-t005:** Categorical parameters (Chi-square test) for 48 h mortality.

Variable	Alive (%)	Dead (%)	*p* Chi2
Encephalopathy	16.9	33.3	*0.0021*
Hepatocellular carcinoma	4.08	8.3	*0.0952*
Portal vein thrombosis	4.4	2.1	0.3037
Gastric varices	7.8	10.4	0.4252
Forrest	40.1	39.6	0.9243
Endoscopy treatment	61.4	56.3	0.3621
CTP class (A/B/C)	20/55.8/18.8	11.5/39.6/44.8	*0.0000*
Cirrhosis etiology			*0.0098*
-Alcohol	72.1	67.7	
-Viral	27.1	32.9	

CTP = Child-Turcotte-Pugh. Statistically significant *p*-values are marked with italicized fonts.

**Table 6 diseases-13-00265-t006:** Numerical parameters and *p*-values (Mann–Whitney test) for 48 h mortality.

Variable Mean (IQR)	Alive	Dead	*p* Mann–Whitney
AGE	59 (51–66)	62 (52–68)	0.0609
Hb	8.38 (6.63–10.12)	7.91 (6.63–9.43)	0.1582
NLR	4.50 (2.71–7.48)	6.05 (4.42–9.57)	*<0.0001*
Platelet count	93,400 (62,610–131,500)	103,700 (73,787.5–151,250)	*0.0368*
Urea	49.5 (34–71.75)	65 (45.75–95.25)	*0.0001*
Creatinine	0.7 (0.62–0.84)	0.82 (0.6775–1.2825)	*<0.0001*
INR	1.54 (1.334–1.81)	1.65 (1.38–2.21)	*0.0111*
Albumin	2.8 (2.4–3.2)	2.5 (2–2.8)	*<0.0001*
Total bilirubin	1.93 (1.2–3.215)	2.555 (1.2775–5.9775)	*0.0071*
Na	136 (132–139)	135 (131–137)	*0.0374*
ALT	35 (23.5–55)	40 (27.75–83.5)	*0.0353*
Systolic blood pressure	120 (105–135)	110 (100–130)	*0.0011*
CTP	9 (8–10)	10 (9–12)	*<0.0001*
CTP-creatinine	9 (8–10)	11 (9–13)	*<0.0001*
GBS	10 (8–12)	11 (8–12)	0.7925
CRS	4 (4–5)	4 (4–5)	0.5811
AIMS65	2 (1–2)	2 (1–2)	0.8954
FRS	6 (5–7)	6 (5–7)	0.9619
mGBS	7 (5–9)	8 (5–9)	0.6112
CCI	5 (4–5)	5 (4–6)	*0.0003*
MELD	15 (12–19)	15 (11–18)	0.1256
MELD-3	33 (27.25–40.75)	32 (25–38)	0.1851
MELD-Na	17 (13–21.98)	16 (11–20.864)	0.1240
UKELD	52 (50–56)	52 (49–55)	0.1599
ALBi	−1.30 (−1.66–−0.87)	−1.25 (−1.70–−0.85)	0.8732
PALBi	−2.50 (−3.29–−1.70)	−2.69 (−3.63–−1.90)	0.1672

NLR = neutrophil-to-lymphocyte ratio; GBS = Glasgow–Blatchford score; mGBS = modified Glasgow-Blatchford score; CRS = clinical Rockall score; FRS = full Rockall score; CCI = Charlson comorbidity index; CTP = Child-Turcotte-Pugh score; CTP-creatinine = Child-Turcotte-Pugh-creatinine score; ALBi = albumin-bilirubin score; PALBi = platelet-albumin-bilirubin score. Statistically significant *p*-values are marked with italicized fonts.

**Table 7 diseases-13-00265-t007:** Logistic model for 48 h mortality.

Variable	DF	Chi-Square (Wald)	Prob > Wald	Chi-Square (LR)	Prob > LR
Age	1	0.1425	0.7058	0.1438	0.7045
NLR	1	1.8644	0.1721	1.8374	0.1752
Platelet	1	0.0035	0.9525	0.0035	0.9524
Creatinine	1	0.3298	0.5657	0.3888	0.5329
INR	1	9.5275	0.0020	10.4284	0.0012
Albumin	1	8.2249	0.0041	9.3848	0.0022
Total bilirubin	1	0.0019	0.9650	0.0019	0.9650
Etiology: alcohol vs. viral	1	0.0005	0.9805	12.5589	0.0004
Hepatocellular carcinoma	1	1.4128	0.2346	1.2260	0.2682

**Table 8 diseases-13-00265-t008:** The correctness of classification (48 h mortality).

From\To	Presumed Alive	Presumed Dead	Total	% Correct
Alive	365	2	367	99.46
Death	20	2	22	9.09
Total	385	4	389	94.34

**Table 9 diseases-13-00265-t009:** Logistic model for 6-week mortality.

Variable	DF	Chi-Square (Wald)	Prob > Wald	Chi-Square (LR)	Prob > LR
Age	1	8.8426	0.0029	9.4360	0.0021
NLR	1	3.7683	0.0522	3.7536	0.0527
Platelet	1	6.5552	0.0105	6.4653	0.0110
Creatinine	1	10.3801	0.0013	12.5999	0.0004
INR	1	4.2073	0.0402	4.3986	0.0360
Albumin	1	15.2731	<0.0001	16.6575	<0.0001
Total bilirubin	1	2.2143	0.1367	2.4047	0.1210
Etiology: alcohol vs. viral	1	3.1231	0.0772	3.30532	0.0691
Hepatocellular carcinoma	1	2.6027	0.1067	2.5729	0.1087

**Table 10 diseases-13-00265-t010:** The correctness of classification (6-week mortality).

From\To	Presumed Alive	Presumed Dead	Total	% Correct
Alive	273	17	290	94.14
Death	58	41	99	41.41
Total	331	58	389	80.72

## Data Availability

The datasets obtained and used during the current study are available from the corresponding author upon request.

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
