# Peer review of "Is There a Role for the Neutrophil-to-Lymphocyte Ratio for Rebleeding and Mortality Risk Prediction in Acute Variceal Bleeding? A Comparative 5-Year Retrospective Study"

_diseases, 2025, doi:10.3390/diseases13080265_

Round 1
Reviewer 1 Report
Comments and Suggestions for Authors
- The predictive model was developed using a retrospective dataset from a single center and no external validation was performed. In my opinion, this limits the generalizability of the findings. Therefore, the authors should perform external validation in an independent cohort as this is crucial to confirm the utility of the proposed simplified model for predicting 6-week mortality.
- The authors state that 415 patients were enrolled, but do not state the inclusion/exclusion criteria. Were patients excluded for any reason (active malignancy, non-cirrhotic portal hypertension)? Was there heterogeneity in treatment protocols over the 5-year period (2017–2021)? Also, there is no mention of standardization of diagnosis or treatment protocols for AVB (timing of endoscopy, use of antibiotics, vasoactive agents), which could introduce confounding effects and affect outcomes such as mortality and rebleeding. I believe that these data and the answers to these questions are very important and the authors must include them in their scientific article.
- Authors claims poor accuracy of all scores in predicting rebleeding, but does not define rebleeding criteria or its timing. Were standardized definitions (Baveno VI criteria) used? Were early and late rebleeding differentiated? Authors must address these issues clearly.
- Although laboratory data such as creatinine, bilirubin, INR, and albumin were used in the model, authors does not specify the timing of these tests in relation to the bleeding event. This is critical, as lab values fluctuate rapidly during acute episodes.
Author Response
RESPONSE TO REVIEWER 1
- The predictive model was developed using a retrospective dataset from a single center, and no external validation was performed. In my opinion, this limits the generalizability of the findings. Therefore, the authors should perform external validation in an independent cohort, as this is crucial to confirm the utility of the proposed simplified model for predicting 6-week mortality.
R: Thank you for your observations. We acknowledge that the lack of external validation is a limitation of our study and may affect the generalizability of the predictive model. Although our study was conducted at a single center and we did not have access to an appropriate control cohort at the time of this analysis to perform validation, it is worth noting that this center is a large regional hospital serving a diverse patient population that reflects the broader national demographic. As such, we believe the findings may still apply to the general population within the country. Nevertheless, we agree that external validation remains essential to confirm the model’s performance in other real-life settings, and this limitation has now been clearly stated in the revised manuscript (Lines 337-350).
- The authors state that 415 patients were enrolled, but do not state the inclusion/exclusion criteria. Were patients excluded for any reason (active malignancy, non-cirrhotic portal hypertension)? Was there heterogeneity in treatment protocols over the 5 years (2017–2021)? Also, there is no mention of standardization of diagnosis or treatment protocols for AVB (timing of endoscopy, use of antibiotics, vasoactive agents), which could introduce confounding effects and affect outcomes such as mortality and rebleeding. I believe that these data and the answers to these questions are very important, and the authors must include them in their scientific article.
R: Thank you for your observation. Inclusion criteria were acute variceal bleeding confirmed by endoscopy in patients with cirrhosis confirmed by typical clinical, laboratory data, and ultrasound signs. Patients aged below 16 years, with non-cirrhotic variceal bleeding, with BCLC stage D hepatocellular carcinoma or stage IV carcinomas with other locations (because of high non-bleeding related mortality), and those with missing data were excluded (Lines 112-117 added).
We used the same therapeutic protocol during the 2017-2021 period, with no heterogeneity, as described in Lines 120-133. All endoscopies were performed during the first 24 hours after admission; the mean interval ± standard deviation between admission and endoscopy was 7.1 ± 6.2, and 66.7% of patients had endoscopy performed in the first 12 hours after admission. In all AVB cases, antibiotic therapy was recommended from admission (intravenous 3rd-generation cephalosporin or Ciprofloxacin). Terlipressin was administered in all patients with known cirrhosis before endoscopy (usually from the Emergency Department); in patients with no diagnosis of cirrhosis before endoscopy, Terlipressin treatment began immediately after endoscopy and was continued in all cases for 5 days. We modified Lines 120-133 to clearly reflect this information.
- The authors claim poor accuracy of all scores in predicting rebleeding, but do not define rebleeding criteria or their timing. Were standardized definitions (Baveno VI criteria) used? Were early and late rebleeding differentiated? Authors must clearly address these issues.
R: Thank you for the observation. We analyzed only early rebleeding, defined by the Baveno VI consensus as a fresh hematemesis or melena or a hemoglobin drop of more than 3g/dl. We had 32 cases of early rebleeding and 10 cases of late rebleeding. We also corrected a typo in Line 133 regarding the Hb drop (3g/dl instead of 2 g/dl).
- Although laboratory data such as creatinine, bilirubin, INR, and albumin were used in the model, the authors do not specify the timing of these tests in relation to the bleeding event. This is critical, as lab values fluctuate rapidly during acute episodes.
R: Your observations were accurate. Laboratory values at admission were used in our study. We add the information in Lines 111-112.
Reviewer 2 Report
Comments and Suggestions for Authors
Dear authors, in order to improve the quality of the article, I recommend the following changes:
The title of the article is too long, so please change it and shorten it.
In the summary, the sentences are long and unclear, so replace the ; sign with a period and shorten the sentences.
Upper gastrointestinal bleeding (upper GIB) has the abbreviation UGIB in the abstract. Please correct this.
Line 40 ); the mortality is ..please correct with point. The mortality is....
Line 46..TIPS..please write full name
Line 51...ICU...please write full name
Line 69..Table S1.Is this correct S1?
Please clearly cite the European guidelines for the treatment of variceal bleeding in the introduction.
Based on the introduction, I conclude that the sentences are too long and contain errors, so it will be necessary to use the English Editing Office service.
Write clearer inclusion and exclusion criteria.
Line 111 ..Hb ..please write full name
It is necessary to write information about corticosteroid treatment regarding its effect on leukocytes. Also, how does dehydration and bleeding affect neutrophils and lymphocytes?
Write a few sentences for Future directions.
Best Regards
Comments on the Quality of English LanguageDear
The sentences are too long, and the authors are not native speakers.
An English Editing office service is needed to make the article more fluent and clear to read.
I also pointed out some mistakes in the review.
Kind Regards
Author Response
RESPONSE TO REVIEWER 2
Comments and Suggestions for Authors
Dear authors, in order to improve the quality of the article, I recommend the following changes:
- The title of the article is too long, so please change it and shorten it.
R: We shortened the title.
- In the summary, the sentences are long and unclear, so replace the ; sign with a period and shorten the sentences.
R: We replaced all of the signs “;” in the abstract, and we shortened the sentences.
- Upper gastrointestinal bleeding (upper GIB) has the abbreviation UGIB in the abstract. Please correct this.
R: We corrected the usage of “GIB” by replacing it with gastrointestinal bleeding when the term was not referred to as upper gastrointestinal bleeding. We removed the abbreviation GIB at the end of the article.
- Line 40; the mortality is... please correct with a point. The mortality is....
R: Corrected.
- Line 46..TIPS: Please write full name
R: We modified Line 47, with the full name for TIPS, and we used the abbreviation after the first use.
- Line 51...ICU...please write full name
R: We modified Line 53.
- Line 69..Table S1. Is this correct, S1?
R: The Table S1 is correct (it may be found in the Supplementary material).
- Please clearly cite the European guidelines for the treatment of variceal bleeding in the introduction.
R: We modified Lines 45-50, and we added the Baveno V and VI consensus, including the definition of early rebleeding.
- Based on the introduction, I conclude that the sentences are too long and contain errors, so it will be necessary to use the English Editing Office service.
R: We performed a rapid English review using MDPI services.
- Write clearer inclusion and exclusion criteria.
R: We added Lines 112-117.
- Line 111 ..Hb ..please write full name
R: Corrected (the new Line is 124).
- It is necessary to write information about corticosteroid treatment regarding its effect on leukocytes. Also, how does dehydration and bleeding affect neutrophils and lymphocytes?
R: NLR levels can be increased by corticosteroids (which increases neutrophils and may decrease lymphocyte counts), whereas dehydration can decrease both neutrophil function and lymphocyte counts. We added Lines 93-96.
- Write a few sentences for Future directions.
R: Future studies analyzing the role of NLR and other simple hematological parameters for the prognosis of variceal and non-variceal bleeding, eventually in scores by combination with other factors, may help to accurately predict both mortality and rebleeding, and also to better manage patients with severe bleeding and hemodynamic instability (Lines 346-350).
Round 2
Reviewer 1 Report
Comments and Suggestions for Authors
Thank you for the specific answers. I have no new comments.
Reviewer 2 Report
Comments and Suggestions for Authors
Dear authors
You have done all corrections according to reviewer 2 comments.
The manuscript is significantly improved.
In this form is suitable for acceptance. Best Regards